# FibroDB: Expression Analysis of Protein-Coding and Long Non-Coding RNA Genes in Fibrosis

**DOI:** 10.3390/ncrna8010013

**Published:** 2022-01-28

**Authors:** Mirolyuba Ilieva, Henry E. Miller, Arav Agarwal, Gabriela K. Paulus, Jens Hedelund Madsen, Alexander J. R. Bishop, Sakari Kauppinen, Shizuka Uchida

**Affiliations:** 1Center for RNA Medicine, Department of Clinical Medicine, Aalborg University, DK-2450 Copenhagen, Denmark; mirolyubailieva@gmail.com (M.I.); jenshm@dcm.aau.dk (J.H.M.); ska@dcm.aau.dk (S.K.); 2Department of Cell Systems and Anatomy, UT Health San Antonio, San Antonio, TX 78229, USA; millerh1@livemail.uthscsa.edu (H.E.M.); bishopa@uthscsa.edu (A.J.R.B.); 3Greehey Children’s Cancer Research Institute, UT Health San Antonio, San Antonio, TX 78229, USA; 4Bioinformatics Research Network, Atlanta, GA 30317, USA; aravagar@umich.edu (A.A.); gabriela.paulus@gmail.com (G.K.P.); 5Language Technologies Institute, Carnegie Mellon University, Pittsburgh, PA 15213, USA; 6Osthus GmbH, 52068 Aachen, Germany; 7May’s Cancer Center, UT Health San Antonio, San Antonio, TX 78229, USA

**Keywords:** fibroblast, fibrosis, gene expression, lncRNA, RNA-seq

## Abstract

Most long non-coding RNAs (lncRNAs) are expressed at lower levels than protein-coding genes and their expression is often restricted to specific cell types, certain time points during development, and various stress and disease conditions, respectively. To revisit this long-held concept, we focused on fibroblasts, a common cell type in various organs and tissues. Using fibroblasts and changes in their expression profiles during fibrosis as a model system, we show that the overall expression level of lncRNA genes is significantly lower than that of protein-coding genes. Furthermore, we identified lncRNA genes whose expression is upregulated during fibrosis. Using dermal fibroblasts as a model, we performed loss-of-function experiments and show that the knockdown of the lncRNAs *LINC00622* and *LINC01711* result in gene expression changes associated with cellular and inflammatory responses, respectively. Since there are no lncRNA databases focused on fibroblasts and fibrosis, we built a web application, FibroDB, to further promote functional and mechanistic studies of fibrotic lncRNAs.

## 1. Introduction

Fibroblasts are the most common cell type in the connective tissue and are found throughout the mammalian body [1,2]. They synthesize and secrete extracellular matrix (ECM) proteins and collagens to maintain the tissue structure. They can be easily isolated from each tissue by simply explanting a piece of tissue in a cell-culture dish as fibroblasts outgrow from such tissue and adhere to the plastic cell-culture dish [3]. In vitro, the morphology of fibroblasts is distinct and described as large, flat, and spindle-shaped cells [4,5]. Upon injury (e.g., an open-wound injury in the skin, myocardial infarction), fibroblasts are activated (called myofibroblasts), which will result in the proliferation of myofibroblasts as well as the deposition of excessive ECM, leading to progressive tissue scarring, fibrosis, and organ dysfunctions [6,7,8]. The characterization of fibroblasts has been a focus of intensive research for many years [9,10,11,12]. However, the main outstanding issue in the field is that there is no single gene/protein marker that can describe fibroblasts since they are a collective term to describe heterogeneous populations of cells [13,14,15].

The use of RNA sequencing (RNA-seq, including single-cell RNA-seq (scRNA-seq)) technology has identified a large number of non-protein-coding (ncRNA) genes. When the length of an ncRNA is longer than 200 nucleotides (nt), this transcript is categorized as a long non-coding RNA (lncRNA) [16,17,18,19]. LncRNAs bind macromolecules (DNA, RNA, and proteins) to regulate various cellular processes, including epigenetics, transcription, post-transcriptional modifications, and translation [20]. Dysregulations of lncRNA expression and functions are associated with many diseases [21]. To accommodate the increased interest to study lncRNAs, a number of lncRNA databases have been introduced [22], which allow for the user to screen for tissue-specific lncRNAs, for example. Yet, there is no lncRNA database focused on fibroblasts currently available.

Due to the increased interest to study fibrosis, a number of RNA-seq data have been deposited in the public domain, such as the Gene Expression Omnibus (GEO) and ArrayExpress. Since most of the published RNA-seq data were analyzed for protein-coding genes but not lncRNAs, analysis of such published RNA-seq data will be a valuable approach to uncover lncRNAs associated with fibrosis. To address this question, we gathered several published RNA-seq data to perform detailed analyses of lncRNAs expressed in fibroblasts and during fibrosis. To facilitate the further search of lncRNAs in fibrosis, we built a web application, FibroDB: https://rnamedicine.shinyapps.io/FibroDB/.

## 2. Results

### 2.1. Tissue-Specific Expressions of Protein-Coding and lncRNA Genes in Human Fibroblasts

The heterogeneity of fibroblasts has been discussed for many years, and there are currently no cell surface markers that identify all types of fibroblasts [23]. To examine the tissue specificity of fibroblasts, expression profiling was performed using microarrays [24,25,26] and RNA-seq [27]. Because microarrays are not optimal for lncRNA expression profiling as most of the microarray platforms are designed only for protein-coding genes, an RNA-seq dataset on fibroblasts from the abdomen, upper gingiva, lung, soft palate, scalp, trachea, and vocal fold was chosen for further analysis (GEO accession number, GSE140523; Ref. [27]). The objective of the original study was to examine differentially expressed genes in vocal-fold fibroblasts compared to fibroblasts from other anatomical locations. Vocal-fold fibroblasts were chosen because they are located in a more stressful tissue environment than those from other body parts, leading to differences in gene expression triggered by mechanical stress in the vocal fold [27]. In the original study, only protein-coding genes were examined, which calls for further analysis of this dataset. When this dataset was analyzed using the latest annotation file provided by the Ensembl database [28] (GRCh38.103), 2405 protein-coding genes (out of 19,796 protein-coding genes without readthrough transcripts registered under the GRCh38.103 annotation file) and 8133 lncRNA genes (out of 16,593 lncRNA genes without readthrough transcripts) were found to be expressed in at least one fibroblast cell line based on the normalized values, counts per million (CPM > 0; Appendix A). Although more than 3.4-fold lncRNA genes in the number of genes are expressed compared to protein-coding genes, the expression of lncRNA genes is significantly lower than those of protein-coding genes (Figure 1A). A previous study showed that lncRNA genes tend to be shorter in length than protein-coding genes [29]. However, lncRNA genes are lowly expressed compared to protein-coding genes even when normalized for their length based on two measurements: Reads per kilobase of transcript per million mapped reads (RPKM) and transcripts per kilobase million (TPM) values (Appendix A). Due to the ongoing debate on the normalization of RNA-seq data and expression values [30], CPM values are used in this study to avoid potential issues related to alternative splicing and isoform length.

When the top 20 highly expressed protein-coding (Figure 1B) and lncRNA (Figure 1C) genes are compared, the expression levels of highly expressed lncRNA genes are 40-fold lower than those of protein-coding genes. The group of highly expressed protein-coding genes comprises collagen (COL1A1, COL1A2, COL3A1, and COL6A3) and structural proteins (ACTB, ACTG1, DCN, FN1, ITGB1, THBS1, and VIM) as expected. Since the 34 fibroblast cell lines were derived from human cadavers of both sexes and various ages [27], XIST (X inactive specific transcript), the master regulator of X chromosome inactivation, is found among the top 20 highly expressed lncRNA genes. Of note, both protein-coding and lncRNA genes are expressed in a cell-line specific manner (Figure 1B,C—heatmaps), although the same procedures were used for tissue explant and culture methodology [27]. This suggests that it will be difficult to identify fibroblast-specific genes simply based on the gene expression.

### 2.2. Gene Expression Changes of Protein-Coding and lncRNA Genes during Pulmonary Fibrosis

Because of constant environmental threats to the lungs (e.g., dusts, viruses) [31,32], fibroblasts in the lungs are exposed to various stress conditions, which are activated to sustain the remodeling of lung tissue upon injury [33]. When this balance goes awry, pulmonary fibrosis occurs, where fibroblasts play an important role [34]. Thus, pulmonary fibroblasts and their gene expression changes are increasingly investigated, especially in the context of severe acute respiratory syndrome coronavirus 2 (SARS-CoV-2) infection [35], which provides a number of RNA-seq data available to be analyzed for lncRNA genes. For example, chronic thromboembolic pulmonary hypertension (CTEPH) is a progressive form of pulmonary hypertension caused by a blockage in the blood vessels in the lungs, leading to pulmonary fibrosis [36,37]. A recent study provided RNA-seq data from fibroblasts isolated from pairs of CTEPH thrombus and pulmonary artery adventitia of the same CTEPH patient (four CTEPH patients in total; GEO accession number, GSE149413; [38]). This study reported transforming growth factor-β (TGF-β) as the central regulator of fibrotic thrombus remodeling via neurophil-mediated inflammation [38]. When a secondary analysis of this RNA-seq dataset was performed, there were 913 up- and 712 down-regulated genes in the thrombus compared to the pulmonary artery adventitia (control) of CTEPH patients at the threshold of 2-fold and *p* < 0.05 (Figure 2A; Appendix A). Of these, 623 up- and 414 down-regulated genes are protein-coding genes, while 221 up- and 221 down-regulated genes encode lncRNAs. However, the expression levels of lncRNA genes are generally lower than those of protein-coding genes in all samples (Figure 2B) as in fibroblasts from other anatomical locations (Figure 1).

TGF-β regulates the growth and differentiation of cells, including fibroblasts [39,40]. In the lungs, TGF-β is a major driver for the development of pulmonary fibrosis [41,42]. To understand the TGF-β-stimulated fibrosis process, a recent study carried out an RNA-seq of the MRC5 lung fibroblastic cell line treated with TGF-β [43]. The authors identified and characterized one lncRNA, *DNM3OS* (DNM3 opposite strand/antisense RNA), from this screening. However, detailed profiling of other lncRNA genes was not provided in this study [43]. To examine the expression changes of lncRNA genes upon the TGF-β stimulation, a secondary analysis of the published RNA-seq dataset (GEO accession number, GSE97829) was performed. We uncovered 1153 up- and 1316 down-regulated protein-coding genes upon the TGF-β treatment, while there are 385 up- and 523 down-regulated lncRNA genes (Figure 3A; Appendix A). Not surprisingly, lncRNA genes are expressed at a much lower level than protein-coding genes (Figure 3B), consistent with our observations in the previous datasets (Figure 1 and Figure 2).

When the two datasets from pulmonary fibroblasts above were compared, there were 85 up-regulated protein-coding and 24 lncRNA genes as well as 76 down-regulated protein-coding and 25 lncRNA genes shared between the two datasets: Fibroblasts from CTEPH patients and MRC5 cells treated with TGF-β (Figure 3C; Appendix A). Gene ontology (GO) analysis of shared differentially expressed protein-coding genes showed that the up-regulated protein-coding genes involved in cell adhesion (GO:0007155) and extracellular matrix organization (GO:0030198) are enriched, which are characteristics of activated fibroblasts, myofibroblasts (Figure 3D). Furthermore, the shared down-regulated protein-coding genes include the GO term for the Wnt signaling pathway (GO:0016055), such as FRZB (frizzled related protein), an extracellular antagonist of Wnt signaling that is known to negatively control the fibrosis in vitro [44]. In summary, the finding of protein-coding and lncRNA genes shared by the two datasets suggest that they are at least in part under the control of TGF-β signaling.

### 2.3. Characterization of TGF-β-Stimulated lncRNAs in Cardiac Fibroblasts

To further characterize the shared up-regulated 24 lncRNAs, another set of RNA-seq data of TGF-β stimulation were analyzed (GEO accession number, GSE123018; [45]). Although this dataset is from cardiac fibroblasts, the use of fibroblasts from two anatomical sources (i.e., lungs and hearts) allows the examination of TGF-β-stimulated lncRNAs in fibroblasts in general. Analysis of RNA-seq data from TGF-β-stimulated cardiac fibroblasts indicates that a limited number of differentially expressed genes (both protein-coding and lncRNA genes) are shared between cardiac and pulmonary fibroblasts, respectively (Figure 4A; Appendix A). Among the up-regulated lncRNAs, there are only three lncRNA genes shared between cardiac and pulmonary fibroblasts based on threshold values of 2-fold and FDR-adjusted *p* < 0.05: AL121749.2, LINC00622, and LINC01711 (Figure 4B).

The lncRNA AL121749.2 (Ensembl Gene ID, ENSG00000287528) is a 1350-nt long lncRNA with four exons, located on the forward strand of chromosome 10 (chr10: 35,641,097–35,647,826). In the Ensembl database, the official description of this lncRNA gene is “novel transcript, antisense to FZD8”. FZD8 (frizzled class receptor 8) is a member of the frizzled gene family that encodes receptors for the Wnt and other signaling pathways [46]. A previous study showed that TGF-β induced fibroblast activation In vitro (MRC5 cells and primary human lung fibroblasts) and bleomycin-induced fibrosis in mice (FZD8-deficient mice) is regulated via FZD8 with WNT5B as the ligand [47], suggesting that the lncRNA AL121749.2 might be a novel lncRNA implicated in this mechanism.

The lncRNA LINC00622 (Ensembl Gene ID, ENSG00000260941; long intergenic non-protein coding RNA 622) is a single exonic lncRNA (1570 nt in length) located upstream (chr1: 119,597,702–119,599,271) of the processed pseudogene, GAPDHP33 (glyceraldehyde 3 phosphate dehydrogenase pseudogene 33). A recent study reported that LINC00622 is found in extracellular vesicles (EVs) of adipose-derived stem cells [48]. Knockdown of LINC00622 in EVs reduced tumor growth in nude mice injected with neuroblastoma cells possibly by regulating the expression of gamma-aminobutyric acid type A receptor subunit rho1 (GABRR1) via the androgen receptor (AR).

Finally, the lncRNA LINC01711 (Ensembl Gene ID, ENSG00000268941; long intergenic non-protein coding RNA 1711) is a single exonic lncRNA (967 nt in length), located on the forward strand of chromosome 20 (chr20: 58,634,772–58,635,738) between the lncRNA gene, APCDD1L-DT (APCDD1L divergent transcript), and the protein-coding gene, STX16 (syntaxin 16). A recent study reported that LINC01711 could be used as a prognostic biomarker for esophageal squamous cell carcinoma along with seven other lncRNAs (AP000487, AC011997, LINC01592, LINC01497, FENDRR, AC087045, and AC137770) [49]. Another study found that LINC01711 is contained in the exosomes of esophageal squamous cell carcinoma (ESCC) [50]. Knockdown of LINC01711 in ESCC cell lines induced apoptosis, inhibited the proliferation, migration, invasion, and cell growth, while the administration of exosome-derived LINC01711 promoted tumor growth in nude mice. Mechanistically, the authors suggested that LINC01711 functions as a microRNA sponge to sequester miR-326, which targets fascin actin-bundling protein 1 (FSCN1). Furthermore, the downstream protein-coding gene, STX16, is shown to interact with the N-terminal region of CFTR (cystic fibrosis transmembrane conductance regulator) in epithelial cells [51], suggesting a possible relation to pulmonary fibrosis. In summary, all three lncRNA genes have one isoform reported in the current annotations by the Ensembl database and have not been studied for their functions in fibroblasts, which calls for detailed mechanistic studies in the context of TGF-β-induced fibrosis.

### 2.4. Loss-of-Function Study of TGF-β-Stimulated lncRNAs in Dermal Fibroblasts

To test whether the above TGF-β-stimulated lncRNA genes have functions during fibrosis, we performed loss-of-function experiments in dermal fibroblasts. The rationale for using fibroblasts derived from yet another anatomical location was to investigate the effects of knockdown of the lncRNA genes in fibroblasts in general. First, the dermal fibroblasts were stimulated with TGF-β1, which resulted in the up-regulation of the major collagen, COL1A1 (Figure 5A). Next, the expression of three lncRNA genes and two up-regulated protein-coding genes was examined. As shown in Figure 5B, LINC00622 and LINC01711 were highly up-regulated upon TGF-β1 stimulation, which confirmed the above selection criteria. Of note, the lncRNA candidate gene, AL121749.2, was not expressed in unstimulated nor in TGF-β1-stimulated dermal fibroblasts.

To uncover the functional importance of specific lncRNAs, it is necessary to perform loss-of-function experiments. When LINC00622 was knocked down using siRNAs and stimulated with TGF-β1 (Figure 5C), there are 608 up- and 319 down-regulated genes when the threshold values of 2-fold and FDR-adjusted *p*-values <0.05 were applied to RNA-seq data (Figure 5D). Among the up-regulated genes, GO terms related to cellular responses (e.g., ions, inflammatory, chemokines) are enriched (Figure 5E), whereas GO terms related to cell proliferation and morphogenesis are enriched among the down-regulated genes (Figure 5F). When the differentially expressed genes were analyzed for KEGG pathways, the mitogen-activated protein kinase (MAPK) signaling pathway (hsa04010) (Figure 5G,H), which plays an important role in the profibrotic processes in various diseases [52,53,54,55], was enriched. These data suggest that LINC00622 is important in the regulation of the TGF-β1 stimulated fibrosis.

Similarly, when the lncRNA, LINC01711, was knocked down using siRNAs and upon stimulation with TGF-β1 (Figure 6A), there were 529 up- and 330 down-regulated genes when the threshold values of 2-fold and FDR-adjusted *p*-values < 0.05 were applied to RNA-seq data (Figure 6B). Among the up-regulated genes, the inflammatory response, especially viral related, and the associated processes, including cytokine signaling pathways, are enriched in GO analysis (Figure 6C), whereas GO terms associated with the extracellular matrix and collagen fibril organizations are enriched in the down-regulated genes (Figure 6D). When the differentially expressed genes were analyzed for KEGG pathways, the cytokine–cytokine receptor interaction (hsa04060) and the TNF signaling pathway (hsa04668) were enriched (Figure 6E,F), suggesting that LINC01711 is involved in regulating inflammatory responses of fibroblasts.

### 2.5. The Resource for Protein-Coding and lncRNA Genes in Fibrosis: FibroDB

As a large number of RNA-seq data were analyzed in this study, focusing especially on lncRNA genes, we have built a web application, FibroDB, to disseminate the obtained information (Figure 7A). FibroDB is an easy-to-use web application that allows end-users to explore expression changes of protein-coding and lncRNA genes using the three most-commonly used normalized expression values (CPM, RPKM, and TPM) (Figure 7B). For each gene, the hyperlink is provided to GeneCards [56] for further information, while each study is linked to the data information provided by the GEO [57]. To visualize comparison of each GEO study included in FibroDB, volcano plots are generated for a specified comparison of two experimental conditions (Figure 7C), allowing users to obtain a global image of the study being analyzed. Furthermore, differentially expressed genes can be compared and analyzed further via visual inspection by heatmaps (Figure 7D) and based on Kyoto Encyclopedia of Genes and Genomes (KEGG) pathways. FibroDB also generates a Venn diagram to compare differentially expressed genes among studies registered in FibroDB (Figure 7E). All data included in FibroDB are downloadable from the Download tab to allow further understanding of gene expression changes in fibroblasts and during fibrosis.

## 3. Discussion

In this study, we find that (i) a large number of lncRNA genes are expressed in fibroblasts and during fibrosis; (ii) compared to protein-coding genes, the overall expression levels of lncRNA genes are much lower in fibroblasts and TGF-β-stimulated fibroblasts; (iii) although TGF-β stimulation is a common mechanism in fibrosis, only very few protein-coding and lncRNA genes share similar profiles between cardiac and pulmonary fibroblasts; and (iv) knockdown of the lncRNAs, *LINC00622* and *LINC01711*, resulted in gene expression changes associated with cellular and inflammatory responses, respectively. However, further functional and mechanistic studies are required to understand the importance of these lncRNAs in fibrosis.

As with any other study, there are limitations to our study. First, all the RNA-seq data analyzed are of poly A-enriched sequencing, although all RNA-seq data are strand-specific sequencing. Thus, it is possible that lncRNAs without poly A tails may have higher expression than those with poly A tails. Second, we only focused on the known lncRNA genes based on the latest annotation provided by the Ensembl database. Thus, it is possible that novel lncRNA genes might have higher expression than those of protein-coding genes. Third, only one time point after the stimulation with TGF-β was investigated. More mechanistic studies are needed with longer stimulation to understand the impact of silencing the candidate lncRNAs identified in this study.

Although it is now common to employ scRNA-seq to assess the heterogeneity of cells [11,58,59,60], such an approach is not suitable for studying the functions of lncRNAs, as it will be difficult to identify minor populations of cells with low expression of lncRNA genes as shown in this study. Thus, we intentionally excluded scRNA-seq data from FibroDB. Since the interest to study fibrosis has increased in recent years, the commands and snakemake [61] pipelines are available via the GitHub repository to allow further analysis of similar RNA-seq data.

There are several databases currently available that include expression profiles of lncRNAs. Most of these databases include the expression profiles derived from RNA-seq data of whole tissues (normal and/or tumors as in the case of C-It-Loci [62], LncBook [63], lncRNAtor [64], LncExpDB [65], RefLnc [66]) and cell lines (LncExpDB [65], lncRNAtor [64], RefLnc [66]), which is not ideal to understand the expression profiles of a certain cell type, especially in normal physiological conditions. To solve this problem, two databases that focus on a specific cell type is available: ANGIOGENES for endothelial cells [67] and RenalDB for cells in kidneys [68]. To the best of our knowledge, FibroDB is the first lncRNA database focused specifically for fibroblasts and during fibrosis. Our FibroDB web application allows the users to quickly search for lncRNAs differentially expressed in several experimental conditions. Furthermore, comparisons among different experimental settings can be carried out to narrow down the list of differentially expressed lncRNAs during fibrosis in different tissues.

## 4. Materials and Methods

### 4.1. RNA-Seq Data Analysis

RNA-seq data were downloaded from the Sequence Read Archive (SRA) database using SRA Toolkit [69]. The data-sets used in this study are indicated in the Results section with the corresponding accession numbers from the Gene Expression Omnibus (GEO) database. FASTQ files were preprocessed with fastp [70] (versions 0.21.0 and 0.22.0) using default settings to perform quality control, trimming of adapters, filtering by quality, and read pruning. After the preprocessing of sequencing reads, STAR [71] (versions 020201 and 2.7.9a) was used to map the reads to the reference genome (GRCh38.103). To calculate counts per million (CPM) values and derive differentially expressed genes, the R package, edgeR [72] (versions 3.30.3 and 3.32.1), was used. False discovery rate (FDR)-adjusted *p*-values were used for further analysis, unless stated otherwise in the text. To derive reads per kilobase of transcript per million mapped reads (RPKM) and transcripts per kilobase million (TPM) values, the R packages, GenomicFeatures [29] and edgeR, were used. The commands and programs used in this study can be found on the GitHub repository (https://github.com/heartlncrna/Analysis_of_FB_Studies).

### 4.2. Data Analysis and Visualization

To plot violin and volcano plots, the R-package, ggplot2 [73], was used after removing genes with zero CPM values. Volcano plots were generated using bioinfokit [74]. To draw heat maps, the MultiExperiment Viewer (MeV) [75] was used. The gene ontology (GO) terms and the Kyoto Encyclopedia of Genes and Genomes (KEGG) pathways were analyzed via the Database for Annotation, Visualization, and Integrated Discovery (DAVID) v6.8 [76,77].

### 4.3. Cell Culture

The dermal fibroblasts were obtained from the NIGMS Human Genetic Cell Repository at the Coriell Institute for Medical Research (Camden, NJ, USA): GM00730, isolated from a 45-year-old apparently healthy white female. The cells were cultured in the growth medium containing Minimum Essential Medium Eagle (MEM, Sigma-Aldrich, Darmstadt, Germany, #M2279) supplemented with 15% fetal bovine serum (Sigma-Aldrich, #F4135), 1% MEM Non-essential Amino Acid Solution (Sigma-Aldrich, #M7145), 1% L-Glutamine solution (Sigma-Aldrich, #G7513), and 1% Penicillin-Streptomycin (Sigma-Aldrich, #P4333). The cells were cultured at 37 °C with 5% CO_2_.

To silence the target lncRNAs, the following MISSION siRNAs (Sigma-Aldrich) were used: *LINC00622* sense GCUUGUUCUCCGAUAGCUA[dT][dT]/antisense UAGCUAUCGGAGAACAAGC[dT][dT]; and *LINC01711* sense GUCUGGAGCCGUUUCUCUC[dT][dT]/antisense GAGAGAAACGGCUCCAGAC[dT][dT]. The control siRNA used was Mission Negative control SIC-002, confidential sequence (Sigma-Aldrich). Transient siRNA transfection (100 nM final concentration) was carried out using RNAiMax (Life Technologies, Carlsbad, CA, USA) according to the manufacturer’s protocol. The samples were collected 72 h after the transfection of siRNAs for the isolation of total RNA.

To induce fibrosis, cells were serum-starved for one day in MEM supplemented with 1% MEM Non-essential Amino Acid Solution (Sigma-Aldrich, #M7145), 1% L-Glutamine solution (Sigma-Aldrich, #G7513), and 1% Penicillin-Streptomycin (Sigma-Aldrich, #P4333) at 37 °C with 5% CO_2_. Then, the media were changed with the growth medium above, which are designated as the control (unstimulated). To stimulate fibrosis, recombinant human TGF-β1 (HEK293 derived) (Proteintech, Manchester, UK, #100-21) was added to the growth medium at the concentration of 5 ng/mL. In the case of siRNA-based knockdown, siRNA was added to the culture medium at the time of medium change. The samples were collected 24 h after the stimulation with TGF-β1 for immunostaining and isolation of total RNA.

### 4.4. Isolation of Total RNA and RT-PCR

The TRIzol Reagent (Thermo Fisher Scientific, Roskilde, Denmark, #15596018) was used to isolate the total RNA from cells and purified following the manufacturer’s protocol. SuperScript IV VILO Master Mix with the ezDNase Enzyme (Thermo Fisher Scientific, #11766500) was used to digest the genomic DNA and reverse transcribe one μg of total RNA for each sample to synthesize the first-strand complementary DNA (cDNA). After reverse transcription, the first-strand cDNA was diluted with DNase/RNase-free water to the concentration of 1 ng/μL. A quantitative reverse transcription polymerase chain reaction (qRT-PCR) reaction was performed with 1 ng of cDNA template per reaction using PowerUp SYBR Green Master Mix (Thermo Fisher Scientific, #A25777) via the QuantStudio 6 Flex Real-Time PCR System (Thermo Fisher Scientific) with the annealing temperature at 60 °C. Relative fold expression was calculated by 2^-DDCt^ using ribosomal protein lateral stalk subunit P0 (*PRLP0*) as an internal control. The primer pairs were designed using Primer3 (http://bioinfo.ut.ee/primer3-0.4.0/, 24 December 2021) [78] and in silico validated with the UCSC In-Silico PCR tool (https://genome.ucsc.edu/cgi-bin/hgPcr, 24 December 2021) before extensive testing by the conventional RT-PCR reaction followed by running the PCR product on an agarose gel to examine for a single band of the expected size for each primer pair. The primer sequences are provided in Appendix A.

### 4.5. Immunocytochemistry

Cells were washed once with ice-cold phosphate-buffered saline (PBS) and fixed with 4% paraformaldehyde (Thermo Fischer Scientific, #28906) in PBS for 10 min. After three washes with ice-cold PBS, cells were permeabilized with 0.1% Triton-X 100 (Sigma-Aldrich, T8787) in PBS for 5 min and washed once with 0.1% Triton-X 100 in PBS. The primary antibody was diluted in PBS with 30% Blocker BSA (Thermo Fisher Scientific, #37525) and 0.1% Triton X-100. The incubation of the primary antibody was conducted at 4 °C for overnight. After three washes with 0.1% Triton-X 100 in PBS, the secondary antibody was diluted in 0.1% Triton-X 100 in PBS to stain at room temperature (RT) for one hour: Goat anti-Rabbit IgG (H + L) Cross-Adsorbed ReadyProbes Secondary Antibody, Alexa Fluor 488 (Thermo Fisher Scientific, #R37116). The stained cells were washed three times with 0.1% Triton-X 100 in PBS. To visualize the nuclei, DAPI (Sigma-Aldrich, #10236276001) staining was performed at room temperature for 5 min. The antibodies used in this study are provided in Appendix A. The stained cells were visualized with the Invitrogen EVOS FL Digital Inverted Fluorescence Microscope (Thermo Fisher Scientific). The obtained images were merged using the Fiji image processing package [79].

### 4.6. RNA-Seq Experiment

Non-directional RNA-seq was performed at Novogene (Cambridge, UK) using the NEB Next Ultra RNA Library Prep Kit for mRNA-seq library preparation followed by sequencing with the Illumina Novaseq 6000 platform with a PE150 sequencing strategy.

### 4.7. FibroDB Web Application

The FibroDB web application is based on the R package, Shiny [80]. The app provides three primary features: (1) Exploration of results (Explore), (2) downloads, and (3) documentation. The Explore tab displays an interactive table, created with the R package, DT (https://github.com/rstudio/DT), containing differential gene expression results from the study selected by the user. Normalized expression values were plotted via the R packages, ggplot2 [73] and plotly [81]. The table and plot are linked, such that when a user selects a row in the table, the plot changes to reflect the expression of the gene on that row. The Explore tab also displays the volcano plot for each study for which differential gene expression could be calculated. The plot is visualized using the R package, ggplot2.

At the Explore page, the Heatmap tab displays an expression heatmap plotted via the pheatmap function from the R package, ComplexHeatmap [82]. Pathway enrichment was calculated using the en-richr function from the R package, enrichR [83,84,85,86], to query over- or under-expressed differentially expressed genes (DEGs) against the KEGG pathway database. The results are visualized in the Pathway analysis tab using the R package, ComplexHeatmap [82]. Finally, the over- or under-expressed genes are compared between studies using the venn.diagram function from the R package, VennDiagram [87], and visualized in the Comparison tab.

The processed datasets from this study are hosted on a public AWS S3 bucket. The Downloads page provides instructions and links to access these data along with verbose descriptions. The Documentation page provides instructions for the usage of the application, rendered as HTML via the R package, R Markdown [88]. FibroDB will be updated twice a year to include the latest publicly available RNA-seq datasets after manual search via the GEO database.

All code used to generate FibroDB is available in the GitHub repository: https://github.com/Bishop-Laboratory/FibroDB.

### 4.8. Statistics

Data are presented as the mean ± S.E.M. unless otherwise indicated. Two-sample, two-tail, heteroscedastic Student’s *t*-test was performed to calculate a *p*-value via Microsoft Excel.

## 5. Conclusions

There are many lncRNA databases currently available [22]. Yet, all these databases catalog a variety of lncRNAs in various tissues, of which some databases are aimed to uncover tissue-specificity and conservation among species [89]. Our FibroDB differs from the existing lncRNA databases by focusing on one cell type (that is, fibroblasts) and their differentiated status, myofibroblasts, which are increasingly being studied in chronic inflammatory diseases. As experimentally demonstrated in this study, part of the contents in FibroDB have been confirmed via expression profiling of fibrotic genes and upon performing loss-of-function experiments for two identified lncRNAs using dermal fibroblasts stimulated with TGF-β1. These validation data will add confidence to the information contained in FibroDB for further biological experiments for end-users to explore the roles of new lncRNAs implicated in fibrosis.

## Figures and Tables

**Figure 1 ncrna-08-00013-f001:**
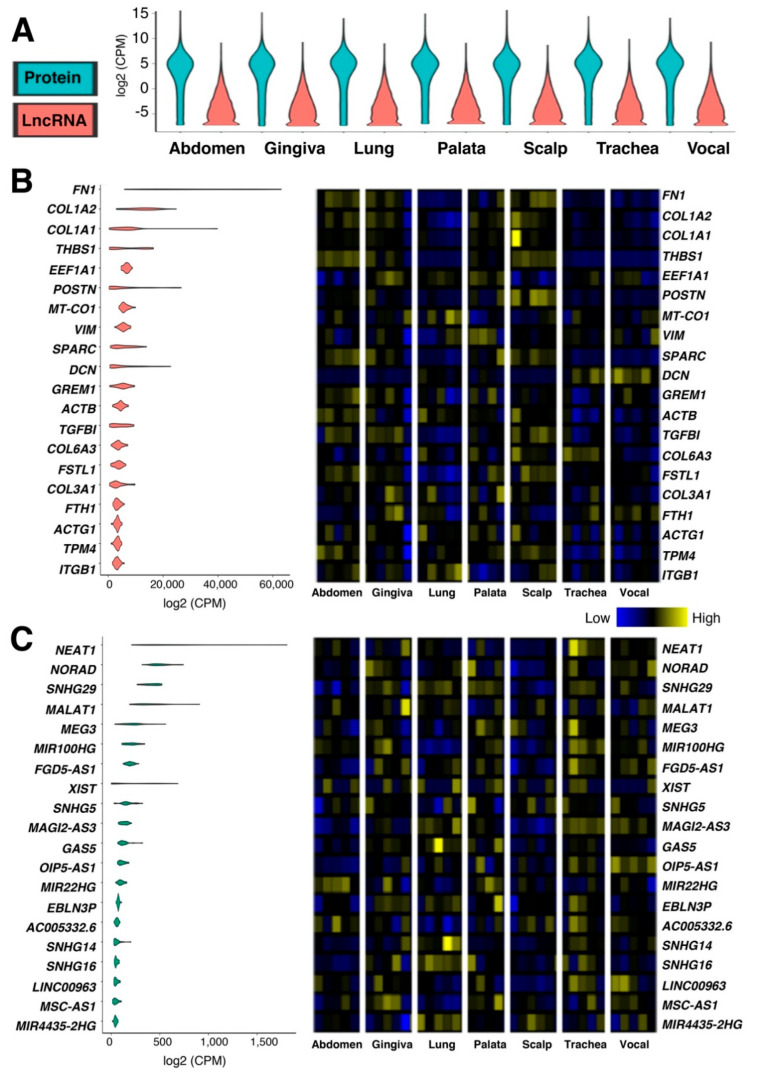
Gene expression of tissue-specific human fibroblasts. (**A**) RNA-seq data of human fibroblast primary cell lines from healthy cadavers isolated from seven anatomical locales: Abdomen, upper gingiva, lung, soft palate, scalp, trachea, and vocal fold. *n* = 5 biological replicates per anatomical locale, except for soft palate, which is *n* = 4 biological replicates. The average CPM (counts per million) values for each anatomical locale were used to draw a violin plot for protein-coding and lncRNA genes based on the annotation (biotype) provided by the Ensembl database (GRCh38.103). (**B**,**C**) Top 20 (**B**) protein-coding and (**C**) lncRNA genes based on the average CPM values of all samples, consisting of *n* = 5 biological replicates per anatomical locale, except for soft palate, which is *n* = 4 biological replicates. The violin plot of each gene (left) is followed by the heat map of gene expression (right) to show the expression differences among isolated body parts. The heatmaps are normalized for CPM values per gene to visualize the sample differences.

**Figure 2 ncrna-08-00013-f002:**
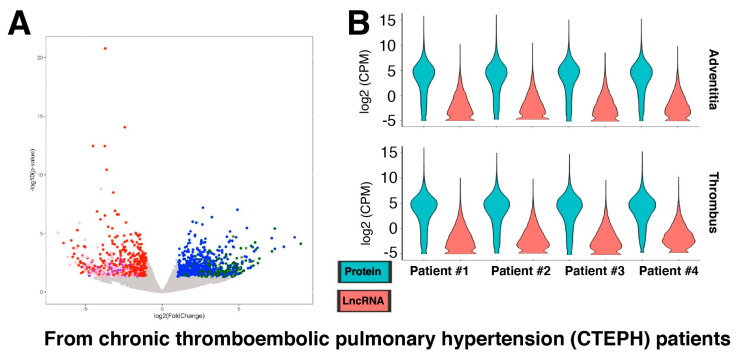
Differential gene expression in fibroblasts from chronic thromboembolic pulmonary hypertension (CTEPH) patients. (**A**) Volcano plot of all genes, including protein-coding and lncRNA genes. The up-regulated protein-coding genes are shown in blue, lncRNA genes in dark green, and other RNA species (based on biotypes provided by the Ensembl database) in olive. The down-regulated protein-coding genes are shown in red, lncRNA genes in pink, and other RNA in purple. With the threshold values of 2-fold and *p* < 0.05, there are 913 up- and 712 down-regulated genes in thrombus compared to pulmonary artery adventitia (control) of CTEPH patients (*n* = 4 biological replicates). (**B**) Violin plots of RNA-seq data of fibroblasts from all four CTEPH patients. The isolation sources of fibroblasts (adventitia and thrombus) are plotted separately as indicated above.

**Figure 3 ncrna-08-00013-f003:**
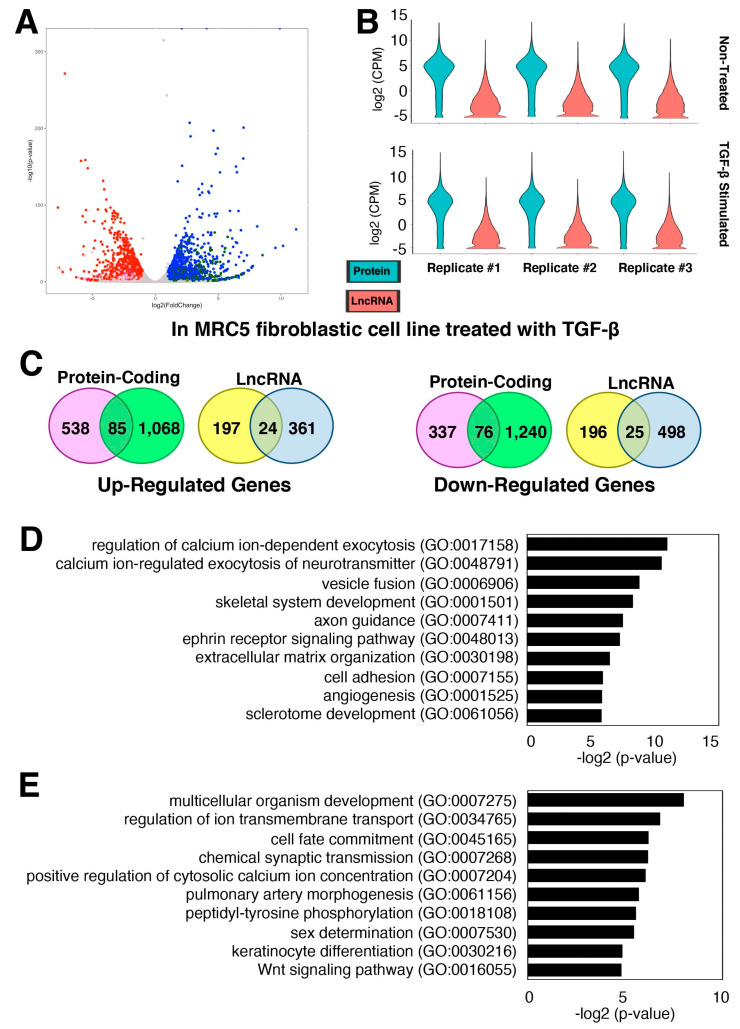
Gene expression changes in activated fibroblasts, myofibroblasts. (**A**,**B**) Differential gene expression in MRC5 fibroblastic cell line treated with TGF-β. (**A**) Volcano plot of all genes, including protein-coding and lncRNA genes. The up-regulated protein-coding genes are shown in blue, lncRNA genes in dark green, and other RNA species (based on biotypes provided by the Ensembl database) in olive. The down-regulated protein-coding genes are shown in red, lncRNA genes in pink, and other RNA in purple. With the threshold values of 2-fold and FDR-adjusted *p* < 0.05, there are 1692 up- and 1957 down-regulated genes in TGF-β-treated MRC5 cells compared to non-treated cells (control) (*n* = 3 technical replicates). (**B**) Violin plots of RNA-seq data plotted separately for each replicate as indicated above. (**C**) Venn diagrams of differentially expressed genes in two datasets: Fibroblasts from CTEPH patients (in pink or yellow) and MRC5 cells treated with TGF-β (in green or light blue). (**D**,**E**) Top 10 enriched GO terms (biological process) for shared (**D**) up- and (**E**) down-regulated protein-coding genes in two datasets.

**Figure 4 ncrna-08-00013-f004:**
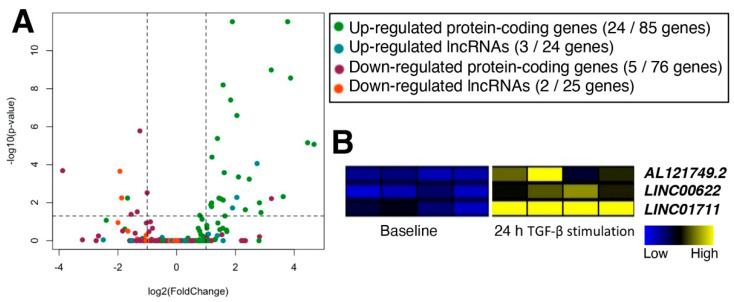
TGF-β-stimulated lncRNAs in cardiac and pulmonary fibroblasts. (**A**) Volcano plot of genes from RNA-seq data of cardiac fibroblasts stimulated with TGF-β for 24 h compared to the CFs before the TGF-β stimulation (baseline). Only genes selected as differentially expressed in pulmonary fibroblasts (CTEPH patients and MRC5 cells treated with TGF-β) were plotted. Each category of genes is colored as shown in the graph label. In each parenthesis, up- and down-regulated protein or lncRNA genes are shown as the number of genes in cardiac fibroblasts followed by the same category of differentially expressed genes in pulmonary fibroblasts. Dashed lines indicate 2-fold and FDR-adjusted *p* < 0.05. According to the original publication, human primary atrial fibroblasts were prepared from atrial biopsies of patients undergoing coronary artery bypass grafting. *n* = 4 patients. (**B**) Heatmap of three up-regulated lncRNAs in all three datasets of cardiac and pulmonary fibroblasts.

**Figure 5 ncrna-08-00013-f005:**
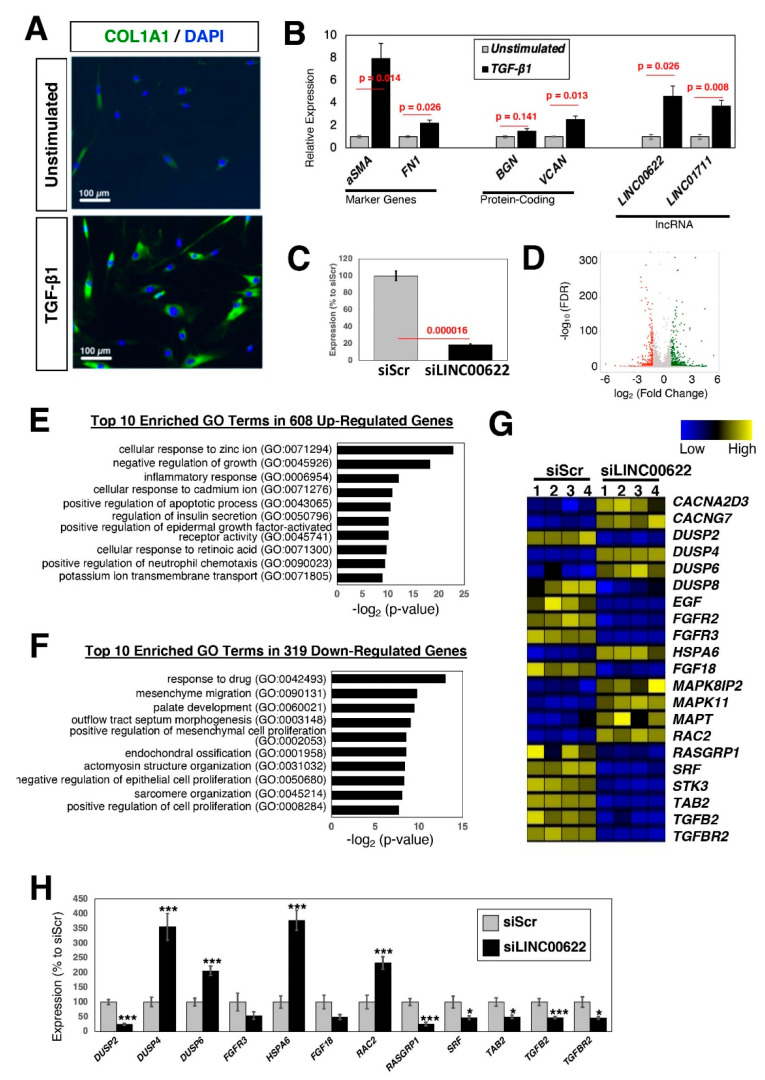
Loss-of-function experiments of the candidate lncRNA, *LINC00622*, in dermal fibroblasts. (**A**) Protein expression of myofibroblast marker, COL1A1. Fibroblasts were stimulated with TGF-β1 (5 ng/mL) for 24 h. The nuclei were counterstained with DAPI. Representative images are shown from more than 3 independent experiments. (**B**) qRT-PCR results of fibrotic marker genes, up-regulated protein-coding, and lncRNA genes. For *AL121749.2*, it is not expressed in both unstimulated and TGF-β1-stimulated dermal fibroblasts. *n* = 4 biological replicates per condition. (**C**) qRT-PCR results upon silencing of *LINC00622*. Twenty-four hours after addition of siRNA to the growth medium, the cells were serum starved for one day and stimulated with TGF-β1 for 24 h in the presence of siRNAs. *n* = 6 biological replicates per condition. (**D**) Volcano plot of RNA-seq data. There are 608 up- and 319 down-regulated genes when the threshold values of 2-fold and FDR-adjusted *p*-values <0.05 were applied. *n* = 4 biological replicates per condition. (**E**,**F**) Top 10 enriched GO terms for (**E**) up- and (**F**) down-regulated genes. (**G**) Heatmap of differentially expressed genes categorized under the KEGG pathway for MAPK. (**H**) qRT-PCR results of differentially expressed genes. *n* = 6 biological replicates per condition. * (*p* < 0.05) and *** (*p* < 0.005).

**Figure 6 ncrna-08-00013-f006:**
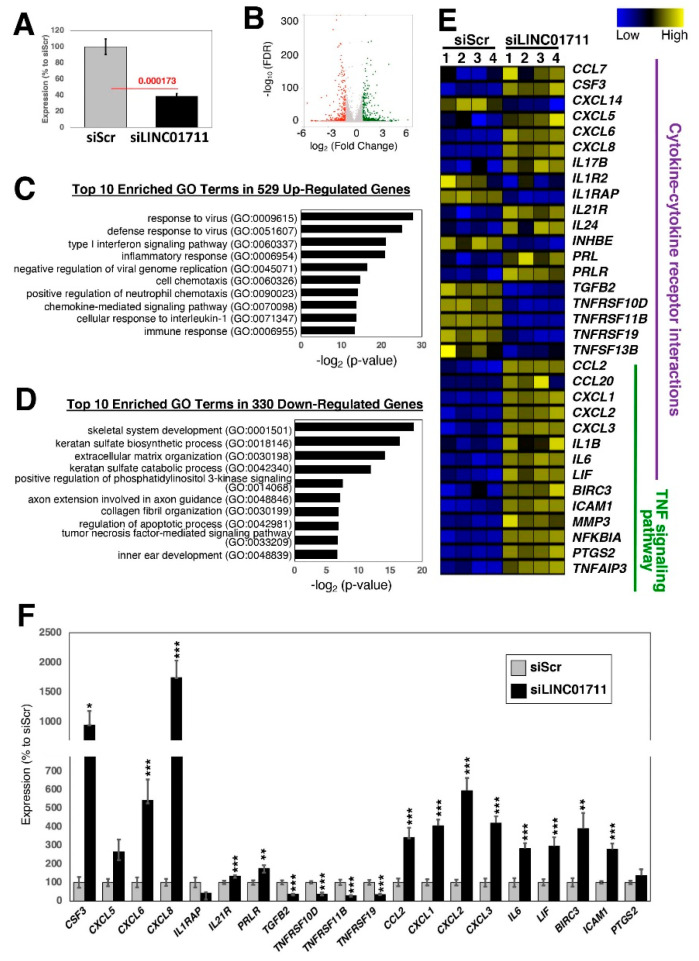
Loss-of-function experiments of the candidate lncRNA, *LINC01711*, in dermal fibroblasts. (**A**) qRT-PCR results upon silencing of *LINC01711*. Twenty-four hours after addition of siRNA to the growth medium, the cells were serum starved for one day and stimulated with TGF-β1 for 24 h in the presence of siRNAs. *n* = 6 biological replicates per condition. (**B**) Volcano plot of RNA-seq data. There are 529 up- and 330 down-regulated genes when the threshold values of 2-fold and FDR-adjusted *p*-values <0.05 were applied. *n* = 4 biological replicates per condition. (**C**,**D**) Top 10 enriched GO terms for (**C**) up- and (**D**) down-regulated genes. (**E**) Heatmap of differentially expressed genes categorized under the KEGG pathways for cytokine–cytokine receptor interaction (hsa04060) and TNF signaling pathway (hsa04668). (**F**) qRT-PCR results of differentially expressed genes. *n* = 6 biological replicates per condition. * (*p* < 0.05), ** (*p* < 0.01), and *** (*p* < 0.005).

**Figure 7 ncrna-08-00013-f007:**
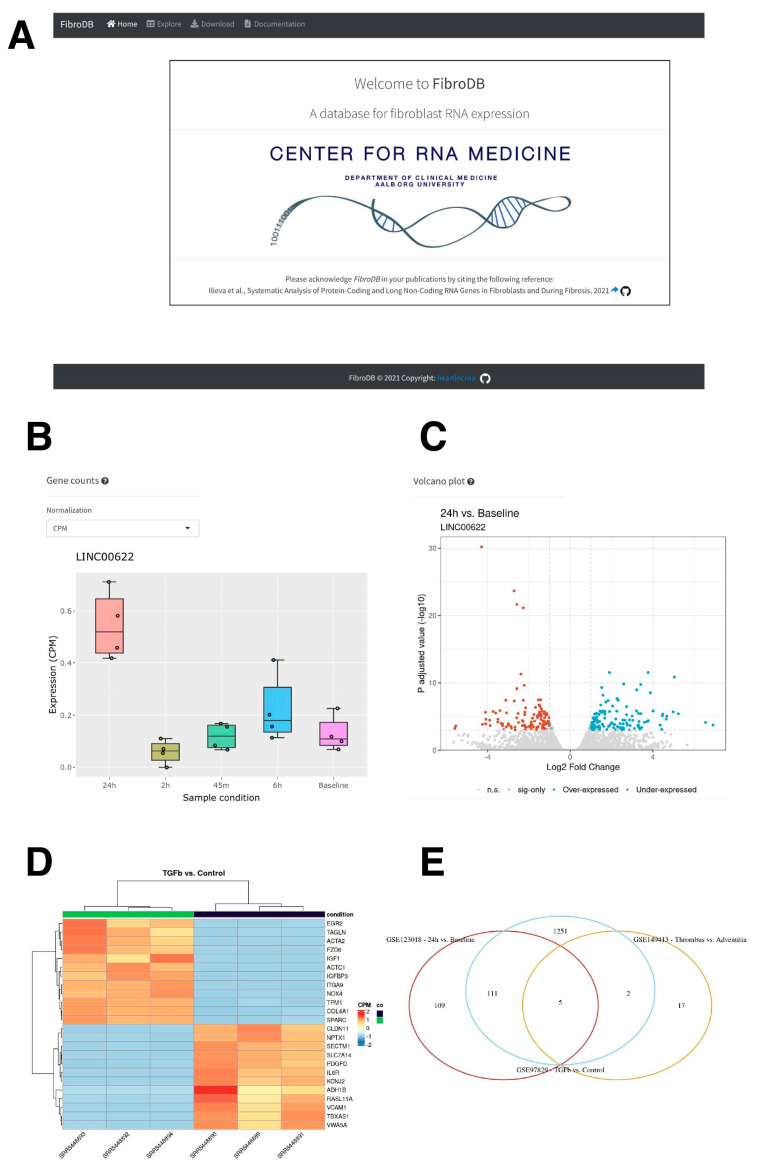
The FibroDB web application. (**A**) The frontpage of FibroDB. (**B**) An example of expression profile of *LINC00622*. The normalized values can be changed from the pull-down menu to select CPM, RPKM, or TPM values. (**C**) An example of volcano plot for the study condition selected. (**D**) An example of heatmap of differentially expressed genes. (**E**) An example of Venn diagram for up-regulated genes among studies registered in FibroDB.

## Data Availability

The Appendix A, commands, and programs used in this study can be found in the GitHub repository: https://github.com/heartlncrna/Analysis_of_FB_Studies). All code used to generate FibroDB is available in the GitHub repository: https://github.com/Bishop-Laboratory/FibroDB.

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
