# Peer review of "FibroDB: Expression Analysis of Protein-Coding and Long Non-Coding RNA Genes in Fibrosis"

_ncrna, 2022, doi:10.3390/ncrna8010013_

Round 1

Reviewer 1 Report

I'm happy with the authors revisions from my comments and also the other reviewers

Author Response

Thank you very much.

Reviewer 2 Report

Authors exemplify expression and potential function of lncRNAs in fibroblasts with fibrosis as model system and additionally offer a novel web resource on fibroblasts lncRNAs namely FibroDB. This work provides insights to the above topic and reusable data to support other researchers in this field via a database. I would add the following comments:

  • The introduction is comprehensibly written but still quite compact and could be extended in terms of current knowledge on exemplary expression and function of lncRNAs in the various cell types and in particular fibroblasts. Furthermore, available lncRNA web resources could be summarized and their shortcomings could be pointed out. (Other databases compared such as lncrnadiseas, RNAcentral, LNCipedia, lncAtlas and more ...)
  • Figure 1 legend lacks indication of 5 or 4 replicates for (B) and (C)
  • The discussion is also little short. I would suggest to add some further information on interpretations, theoretical + practical implications, and possible short-comings and limitations of all results presented.
  • Line 481: Readers may be confused about the conclusion "There are many lncRNA databases currently available" but finding the reference for this statement only after. Maybe you could add 80 to this sentence and leave 81 to the next.
  • The Methods/Materials section (4.3,4.4+) does not specify all timing information such as when cells were collected for analysis or incubation duration of treatments.
  • In regard to the database, options for adding future data (by the host as well as possibilities by third parties even if manual validation is necessary), rendering the option to the database to grow, could be described.
  • The idea of this speciifc database is promising. The tool could be extended to a more general use for other data sets and possibly other diseases with e.g. GEO lncRNA-Seq data as well? Researchers may highly benefit from additional documentation on github!

Round 2

Reviewer 2 Report

Thank you for your amendments, I still have the following comments:

You referenced other resources for lncRNA analysis and mention the difference of focusing fibroblasts. Still, readers may also benefit from learning more about other differences and comparison to existing databases.

Could you please add some more details on differences and/or similarties to other lncRNA databases? F.i. What did you learn from other databases that you could use already in yours?

Why don’t you refer to your existing database and highlight common features as well as peculiarities?

Author Response

Thank you very much for your further comment. The discussion is modified as follows:

“There are several databases currently available that include expression profiles of lncRNAs. Most of these databases include the expression profiles derived from RNA-seq data of whole tissues (normal and/or tumors as in the case of C-It-Loci [63], LncBook [64], lncRNAtor [65] , LncExpDB [66], RefLnc [67]) and cell lines (LncExpDB [66], lncRNAtor [65] , RefLnc [67]), which is not ideal to understand the expression profiles of a certain cell type, especially in normal physiological conditions. To solve this problem, two databases that focus on a specific cell type is available: ANGIOGENES for endothelial cells [68] and RenalDB for cells in kidneys [69]. To best of our knowledge, FibroDB is the first lncRNA database focused specifically for fibroblasts and during fibrosis. Our FibroDB web ap-plication allows the users to quickly search for lncRNAs differentially expressed in several experimental conditions. Furthermore, comparisons among different experimental set-tings can be carried out to narrow down the list of differentially expressed lncRNAs during fibrosis in different tissues.”

This manuscript is a resubmission of an earlier submission. The following is a list of the peer review reports and author responses from that submission.

Round 1

Reviewer 1 Report

Ilieva et al describe FibroDB, an online resource and database specific to the expression profiles of lncRNA and protein-coding genes in fibroblasts.

Fibroblasts are an especially important cell type to study in respiratory diseases such as COPD and pulmonary fibrosis as well as in other tissues such as the heart.

The authors have performed a reanalysis of several publicly available RNA-seq datasets, the novelty being a focus on lncRNAs which was not performed in the original studies. The results have then been made in a web accessible resource as well as downloadable processed data. Code used is also made available via GitHub. This is likely to be very useful to the wider scientific community with an interest in fibroblasts, and especially where this overlaps with lncRNAs.

Minor Comments:

The volcano plots (Fig 2A, Fig3A, Fig4A) it would be useful to be able to visualise the lncRNAs and the protein-coding genes, perhaps with different colour shading (e.g. light & dark green). This would highlight the different distribution of logFold change between lncRNA and protein-coding.

Figure 3C - there are four colours used in the Venn diagrams which are not explained in the legend

It wasn't clear why the 3 lncRNAs in Fig4 were selected (only 3 with a sig adjusted p.value? by logFC?), this criteria should be included in the text.

Ln350, no details of what fastp was used for. I assume trimming, but this isn't stated.

Ln363, Zenodo is not related to Ref 65, or volcano plots

Reviewer 2 Report

The authors proposed in this original article a database and a GUI allowing users to explore lnc-RNA expression. The tool allows the representation of single gene expression or volcano plot within a few mouse clicks. The database is well documented and the interface is clean and reactive. The backend at first glance seems clear and well written. But there are major issues in the proposed concept:

  • functions proposed in the GUI are very limited
  • no possibility to perform downstream analysis such as expression correlation or common gene regulation
  • adding new dataset is not well documented; future data addition and their reviewing is unclear
  • description of experiments, searching / comparing / merging dataset is not implemented in the tool

Moreover, the authors used for the first version of the database a set of preliminary experiments which were not designed to answer the formulated hypothesis. As the authors mentioned in the discussion, polyA enrichment was performed to enrich and sequence mRNA. Their observation and conclusion on the abundance of non-coding RNA when compared to mRNA is debatable. Similarly, lncRNA response of cells treated with TGF is not performed using total RNA-sequencing and therefore would need to be compared in detail to literature and validated using a second more specific method.

The part on the loss of function is interesting. But it needs to be validated due to the experimental design of the sequencing.

With the highlight being focused on the database, the loss of function part is not enough to make the study relevant. Moreover, the database still needs major improvements which are not realistic in a revision.